# Monocular Human-Object Reconstruction in the Wild

## ABSTRACT

Learning the prior knowledge of the 3D human-object spatial relation is crucial for reconstructing human-object interaction from images and understanding how humans interact with objects in 3D space. Previous works learn this prior from datasets collected in controlled environments, but due to the diversity of domains, they struggle to generalize to real-world scenarios. To overcome this limitation, we present a 2D-supervised method that learns the 3D human-object spatial relation prior purely from 2D images in the wild. Our method utilizes a flow-based neural network to learn the prior distribution of the 2D human-object keypoint layout and viewports for each image in the dataset. The effectiveness of the prior learned from 2D images is demonstrated on the human-object reconstruction task by applying the prior to tune the relative pose between the human and the object during the post-optimization stage. To validate and benchmark our method on in-the-wild images, we collect the WildHOI dataset from the YouTube website, which consists of various interactions with 8 objects in real-world scenarios. We conduct the experiments on the indoor BEHAVE dataset and the outdoor WildHOI dataset. The results show that our method achieves almost comparable performance with fully 3D supervised methods on the BEHAVE dataset, even if we have only utilized the 2D layout information, and outperforms previous methods in terms of generality and interaction diversity on in-the-wild images.[1]

## CCS CONCEPTS

• **Computing methodologies** → **Reconstruction**; *Reconstruction*; • **Human-centered computing**;

## KEYWORDS

Human-Object Interaction Reconstruction, 3D Computer Vision

## 1 INTRODUCTION

Human-object interaction reconstruction from a single-view image aims at recovering the 3D information of the human-object pair with a monocular image as input, which is a hybridized task that combines human mesh recovery[15], object shape reconstruction[23], object 6D pose estimation[19] and human-object spatial relation modeling[13]. This task is one of the fundamental problems in 3D

---

[1]We will release the code and the dataset for research purposes.

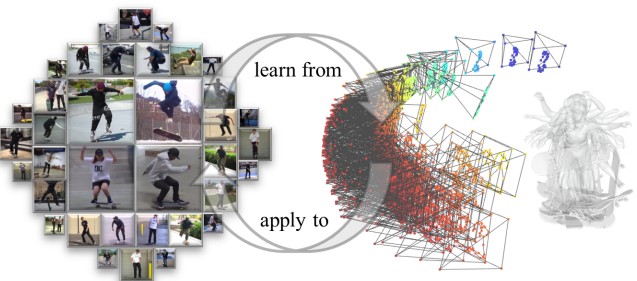

| Unlimited Images in the Wild | Instance-Level Prior |
|---|---|

**Figure 1: In this work, we aim at learning instance-level human-object spatial relation prior from unlimited images in the wild. To accomplish this, we utilize the normalizing flow to learn the view distribution and the 2D human-object keypoints layout on each image plane. The spatial relation prior is then applied to real-world images under the monocular human-object reconstruction setting.**

computer vision and robotics, with potential applications in augmented reality, object manipulation, human behavior imitation, and human activity understanding.

Reconstructing the human and the object jointly is challenging due to the diversity and the complexity of interaction between the human and the object. To address this challenge, recent researchers have proposed various approaches that narrow down the possible range space of the spatial relation between the human and the object by utilizing the prior from commonsense knowledge[32], language model[26] or manually collected dataset[28]. The source from which the priors are acquired significantly constrains the scope within which these methods could be applied. For example, the approach that relies on commonsense knowledge may struggle with uncommon interaction types or novel objects that are not well-represented in the manually crafted rules. Similarly, the approach based on the large language model is limited by the expression capacity of the language model. Moreover, it is also challenging to transfer the highly symbolic language to the 3D real-world interaction prior. The approaches that rely on labor-intensive handcrafted datasets face the same limitation as well. A lightweight process for acquiring human-object spatial relation priors is crucial for the method to handle various interactions and novel objects.

In this work, we introduce a lightweight method that learns the human-object spatial relation priors directly from 2D images collected from the Internet. The key idea behind our method is that the Internet itself is naturally a huge multi-view capturing system, where each individual observes the real world from different perspectives, records images or videos of their surroundings, and uploads them to share their experiences. This results in a huge repository of resources that records various individuals interacting with different objects in different ways. By utilizing this vast amount of data, our method can automatically learn diverse human-object spatial relation priors in a lightweight and efficient manner.

Specifically, our method utilizes a flow-based neural network to learn the distribution of viewports and their corresponding 2D human-object layout in each viewport for the target image. Based on the prior learned from 2D images, we design a scoring function to evaluate the geometric consistency of 3D human-object spatial relations by synthesizing the spatial coherence of the 2D projected human-object keypoints in different image planes. In the optimization stage, we tune the 3D spatial relation between the human and the object by maximizing this scoring function. To prevent the floating phenomenon, we further introduce the contact loss to draw closer the points in the contact candidate regions which are acquired by averaging the occlusion regions of different images. The advantage of our method is that it does not require any manually handcrafted priors or 3D annotations of human-object spatial relation for training, which makes our method scalable to a wide variety of object categories and scenarios. By utilizing the wealth of data available on the Internet, our method can learn how to interact with novel objects which makes it possible to generalize to new unseen objects without the need for manual intervention.

In our experiments, we demonstrate the effectiveness of our method on the BEHAVE dataset by comparing it with 3D-supervised approaches. To validate and benchmark our method on in-the-wild images, we collected a dataset from the YouTube website called WildHOI, which includes various interactions with 8 different object categories such as baseball bat, skateboard, cello, and so on. We manually annotate a small test dataset that contains about 2500 images, with each image containing SMPL pseudo-ground-truth and object 6D pose labels. We then use our method to learn human-object spatial relation prior from the 2D images in the WildHOI dataset and evaluate its performance on the test dataset. Both numerical results and human evaluation show robustness and generalization on in-the-wild images. We summarize our contributions as:

- We present a 2D-supervised approach that learns the spatial relationship prior between humans and objects exclusively from in-the-wild 2D images, without the need for any 3D annotations of humans and objects.
- We demonstrate the effectiveness of integrating prior knowledge derived from 2D images into the post-optimization phase of the human-object reconstruction task. Additionally, we illustrate the efficacy of generating approximate contact maps by averaging occlusion maps from multiple images
- We developed the outdoor WildHOI dataset, which captures a wide variety of real-world interactions in uncontrolled environments. Our evaluation demonstrates the robustness and effectiveness of our method in these complex, in-the-wild images.

## 2 RELATED WORK

**Human-Object Interaction Prior.** The prior knowledge of the spatial relation between the human and the object is vital for human-object reconstruction. Previous approaches explore the usage of the prior from commonsence knowledge[32], language model[26], manually collected datset[28] and synthesized images[11]. PHOSA [32] introduces an optimization-based method that leverages the prior knowledge of the object size and contact parts under interaction to reduce the space of likely 3D spatial configurations. Observing

the commonsense knowledge used in PHOSA requires manual annotations on pairs of bodies and objects, Wang *et al.* [26] utilize the commonsense knowledge from large language models (such as GPT-3) to improve the scalability and the generalizability towards interaction types and object categories. The prior created by human rules or extracted from the large language model, limited by its expressive capability, cannot accurately describe the diverse types of interactions between the human and the object. To fill the blank of 3D full-body human-object interaction dataset, Bhatnagar *et al.*[1] present BEHAVE dataset that captures 8 subjects performing a wide range of interactions with 20 common objects. Xie *et al.*[28] further utilizes this dataset to develop a method named CHORE that learns strong human-object spatial arrangement priors from the BEHAVE dataset. This 3D supervised method significantly outperforms previous optimization-based methods, but its generalizability and scalability are limited by the data where the prior learned from and the significant efforts required to collect the large-scale 3D human-object interaction dataset. More recently, Han *et al.* present a self-supervised method to learn the spatial commonsense of diverse human-object interaction from synthesized images produced by a text-conditional generative model. Their method can be applied to arbitrary object categories without any human annotations. Based on these works, we introduce a novel 2D-supervised method that learns human-object spatial relation prior directly from 2D images.

**3D Prior Learning with 2D Supervision.** When obtaining accurate 3D annotations at scale is expensive and intractable, the methods that rely on 2D supervision are more promising. These methods leverage existing 2D annotated data such as 2D keypoints, masks, or bounding boxes to train models for human pose estimation[5, 31], 3D scene reconstruction[10, 22] or hand-object reconstruction[24]. One common way to eliminate the need for 3D supervision is to use the differentiable renderer to project 3D on 2D with different views and apply 2D supervision on these views. This paradigm allows the model to learn 3D information indirectly through the 2D annotations, making it more feasible to train the 3D models with less manual annotation effort. Inspired by these existing works, we make the first attempt to learn the instance-level human-object spatial relation prior without the use of 3D annotations.

**3D Datasets in the Wild.** Due to the scarcity of 3D annotations, it is very challenging to perform 3D reconstruction in diverse real-world scenarios. To address this challenge, researchers are actively working to develop new algorithms and collect new datasets in the fields of object shape reconstruction, human mesh recovery, and hand-object interaction reconstruction. Some studies [4, 14, 20, 21] develop automated pseudo-annotation pipelines to build large-scale datasets from online sources. which significantly benefits the learning-based methods to handle diverse and natural scenarios. While other approaches[7, 9, 12, 27] use computer graphics techniques to generate synthetic 3D datasets with high-quality annotations, eliminating the need for real-world data collection. By combining these real-world and synthetic datasets, more robust and powerful 3D reconstruction models such as the one proposed in [3], can be trained and can generalize well to various real-world scenarios. Toward the same goal, we collect the WildHOI dataset from the Internet to advance the research community on full-body human-object reconstruction.

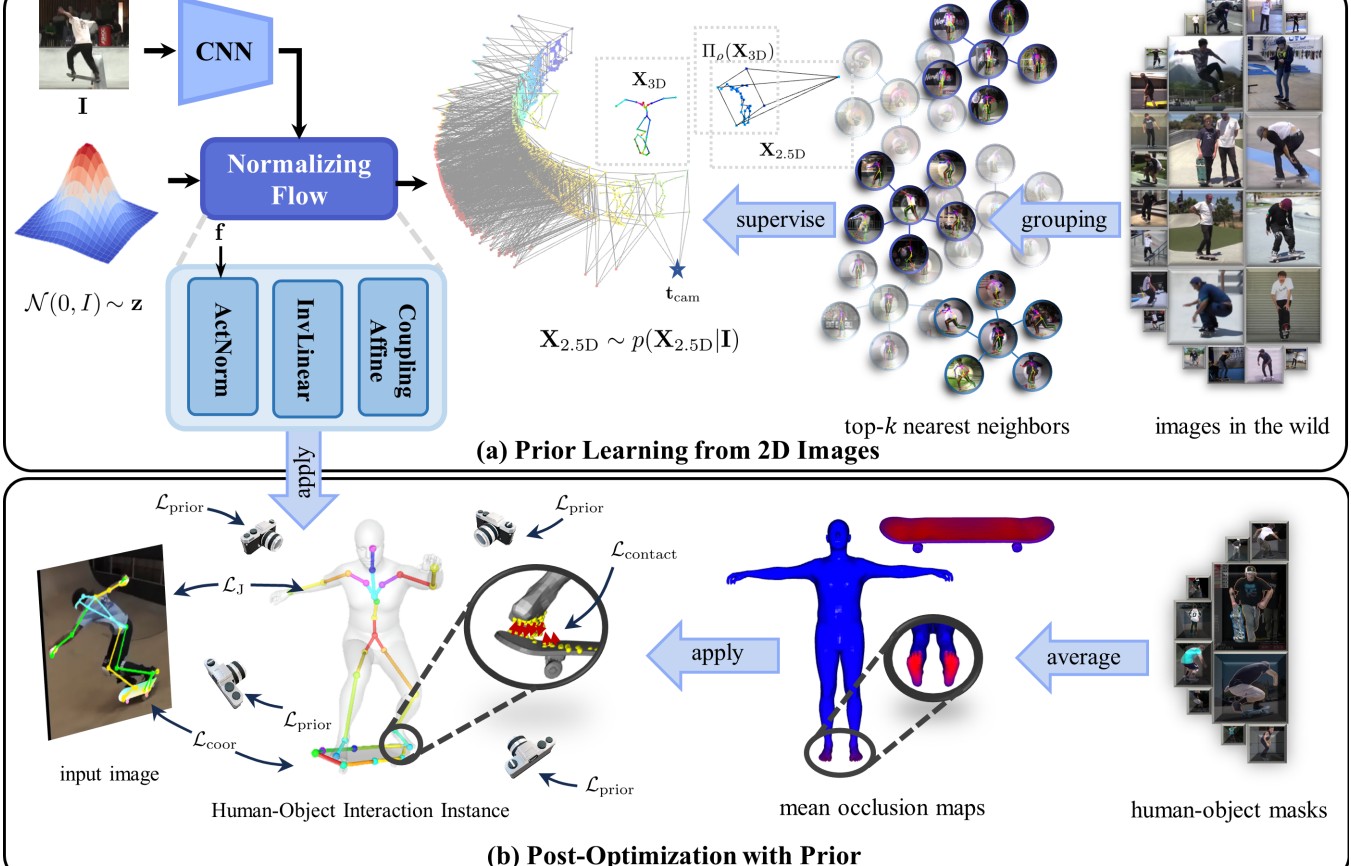

**Figure 2: The main pipeline of our method. We utilize the normalizing flow to learn the distribution of the 2D human-object keypoints in each image plane from vast images in the wild. The normalizing flow takes the input image I as the condition to transform the noize z from Gaussian distribution to the 2.5D keypoints $X_{2.5D}$ which is intermediate representation combining the view pose $\rho$ and the 2D human-object keypoint layout $\Pi_\rho(X_{3D})$. To train this conditioned normalizing flow, we collect a bunch of images from the Internet and group these images together based on the geometry consistency of the 2D human-object keypoints in each view. Then we incorporate the prior learned from 2D images into the post-optimization process. In the post-optimization stage, we project the 3D human-object keypoints onto different image planes of the virtual cameras to ensure the reconstructed results seem coherently observed from other views. Besides, we use the mean occlusion maps that are obtained by averaging the occlusion maps in the images to compute the contact loss. Our method is supervised without using any 3D annotations or commonsense knowledge of the spatial relation between the human and the object.**

## 3 METHOD

**Problem Formulation.** Monocular human-object reconstruction aims at recovering the 3D information $X_{3D}$ [2] of the human and the object given an input image $I \in \mathbb{R}^{h \times w \times 3}$. In order to avoid the ambiguity caused by mutual occlusion between the human and the object in the monocular reconstruction setting, it is more proper to model it as the probability density prediction instead of unimodality estimation. To learn the distribution $p(X_{3D}|I)$ from dataset, the learning-based methods need the 3D annotations for each image. However, due to the high cost of obtaining 3D annotations, it is

difficult to collect a 3D human-object dataset at scale, especially for in-the-wild scenarios. Therefore, these 3D-supervised methods are limited by the distribution of the training dataset, making it difficult to generalize to natural scenes with high diversity. The information about human-object interaction in natural scenes is mostly presented in the form of 2D images or videos, which are easier to be collected from the Internet. Based on this observation, we introduce a method that learns the prior knowledge of 3D human-object spatial relations from large-scale 2D images. To achieve this, we define the scoring function of $X_{3D}$ in the image $I$ as

$$S(X_{3D}|I) = \int_{\rho \sim \varrho} p(\Pi_\rho(X_{3D}), \rho|I)d\rho, \qquad (1)$$

---

[2]The 3D information $X_{3D}$ can be in the form of the 3D point cloud, the parameters of parametric mesh model or any other 3D representation depending on the specific task. Here, we discuss the general case.

where $\rho$ is the pose of the camera, $\Pi_\rho$ is the perspective projection function of the camera under the pose $\rho$, $\varrho$ is the distribution of the camera pose. In defination (1), the 3D information $X_{3D}$ is projected onto different image planes to obtain $\Pi_\rho(X_{3D})$. The score of $X_{3D}$ is obtained by synthesizing the distribution of 2D information from different viewports, which is treated as the approximate to the original 3D density distribution $p(X_{3D}|I)$. In this formulation, the goal becomes to learn the distribution of $p(\Pi_\rho(X_{3D}), \rho|I)$ to approximate the original probability density $p(X_{3D}|I)$.

The content of this section is organized as follows. In section 3.1, we show the representation of $X_{3D}$ and introduce the intermediate representation to bridge the 3D keypoints $X_{3D}$ and the 2D projection $\Pi_\rho(X_{3D})$ under the sparse keypoint representation. In section 3.2, we show how to model and learn the distribution $p(\Pi_\rho(X_{3D}), \rho|I)$ from vast 2D images. In section 3.3, we show how to deploy the scoring function $S(X_{3D}|I)$ to tune the relative pose between the human and the object during post-optimization of human-object reconstruction pipeline.

## 3.1 2D-3D projection

**Human-Object Keypoints.** The raw images captured from the real world contain rich color and geometry information about the arrangement of the human and the object on the 2D image plane. There are several ways to encode the geometry information of the human and the object from unstructured images, such as masks, sparse keypoints and dense coordinates. To make a good balance between computation efficiency and geometry informality, we choose to use the sparse keypoints to represent the human and the object. For the human, we use the joints in the SMPL model as the keypoints. In the SMPL model coordinate system, the keypoints of human $X_{3D}^{SMPL} \in \mathbb{R}^{22 \times 3}$ can be computed as follows:

$$X_{3D}^{SMPL} = \mathcal{J}\mathcal{M}(\boldsymbol{\beta}, \boldsymbol{\theta}), \tag{2}$$

where $\mathcal{M}$ is the blending function which maps the shape parameter $\boldsymbol{\beta} \in \mathbb{R}^{10}$ and the pose parameter $\boldsymbol{\theta} \in \mathbb{R}^{63}$ to the 6890 vertices of the SMPL model and $\mathcal{J} \in \mathbb{R}^{22 \times 6890}$ is the joints weighting matrix. For the object, the keypoints are manually selected from the vertex on the surface of the object mesh model, which is based on the geometric characteristic of the object shape. Denote the localization of the 3D object keypoints under the object local coordinate system as $\hat{X}_{3D}^{object} \in \mathbb{R}^{t \times 3}$, where $t$ is the number of the object keypoints. Under the SMPL local coordinate system, the localization of the object keypoints is computed by

$$X_{3D}^{object} = s\hat{X}_{3D}^{object}R^T + t, \tag{3}$$

where $s$ is the size of the object and $R, t$ is the 6D pose of the object under the SMPL coordinate system. The coordinates of joints of SMPL and the selected object keypoints are concatenated together to get the representation for the 3D human-object spatial arrangement

$$X_{3D} = \begin{pmatrix} X_{3D}^{SMPL} \\ X_{3D}^{object} \end{pmatrix} \in \mathbb{R}^{n \times 3}. \tag{4}$$

In this keypoint-based represetation, the parameters $\{\boldsymbol{\beta}, \boldsymbol{\theta}, R, t, s\}$ are transformed into sparse keypoints $X_{3D}$ under the SMPL local coordinate system.

**The Bridge between 2D and 3D.** Under the perspective camera model, any point $x_{3D} \in X_{3D}$ is projected onto the image plane obtaining the 2D centered coordinates $(u, v)$. The projection relationship between the two is determined by the following equation.

$$\lambda \begin{pmatrix} u \\ v \\ f \end{pmatrix} = R_{SMPL}x_{3D} + t_{SMPL}, \tag{5}$$

where $f$ is the focal length of the camera, $R_{SMPL}$ and $t_{SMPL}$ is the pose of the SMPL model under the camera coordinate system, $\lambda$ is the depth scale. To smplify the notation in following equation, let $x_{2D} = (u, v, f)^T, R_{cam} = R_{SMPL}^{-1}, t_{cam} = -R_{SMPL}^{-1}t_{SMPL}$. Rearrange the terms and normalize both sides, we get

$$\frac{R_{cam}x_{2D}}{\|R_{cam}x_{2D}\|} = \frac{x_{3D} - t_{cam}}{\|x_{3D} - t_{cam}\|}. \tag{6}$$

Equation (6) builds the relation between the 3D keypoints $x_{3D}$ and the coordinate $(u, v)$ on the image plane. Given the observation point $(u, v)$ in the image plane and the pose of the camera $\{R_{cam}, t_{cam}\}$, the corresponding 3D point $x_{3D}$ lies on th ray with the direction $d = \frac{R_{cam}x_{2D}}{\|R_{cam}x_{2D}\|}$ staring from $t_{cam}$. Compute the direction vector $d$ for each point in 3D human-object keypoint $X_{3D}$ according to the right side of equation (6) and concatenate them with $t_{cam}$ obtaining the intermediate representation

$$X_{2.5D} = \left(d_1^T, d_2^T, \ldots, d_n^T, t_{cam}^T\right)^T \in \mathbb{R}^{(n+1) \times 3}, \tag{7}$$

which is the bridge between the 3D keypoints $X_{3D}$ and the projected coordinates $\Pi_\rho(X_{3D})$ with the camera pose $\rho = \{R_{cam}, t_{cam}\}$.

## 3.2 Prior Learning with Normalizing Flow

Because of the scarcity of the 3D annotations for $X_{3D}$, we decompose its distribution into $p(\Pi_\rho(X_{3D}), \rho|I)$ with different viewports that is easy to be acquired from vast images in the Internet. Each pair $\{\Pi_\rho(X_{3D}), \rho\}$ is combined to get the intermediate representation $X_{2.5D}$ according to equation (6)(7). Then the normalizing flow[25] is employed to model the desity distribution $p(\Pi_\rho(X_{3D}), \rho|I)$.

**The Structure of the Normalizing Flow.** The nomalizing flow transform the measurements $X_{2.5D}$ to a sample $z$ in the gaussian distribution with $f$ as the condition, i.e.

$$z = \mathcal{F}(X_{2.5D}; f), \tag{8}$$

where $f$ is the visual feature extracted from the input image $I$. The structure of the normalizing flow $\mathcal{F}$ is constructed using actnorm layer, invertible $1 \times 1$ convolution layer, and the affine coupling layer shown in [16]. The density distribution of $X_{2.5D}$ for the image $I$ is given by

$$\log p(X_{2.5D}|I) = \log q(z) + \left|\det \frac{\partial \mathcal{F}}{\partial X_{2.5D}}\right|, \tag{9}$$

where $q$ is the density function of the Gaussian distribution.

**The Training Process of the Normalizing Flow.** The training objective of the normalizing flow is minimizing the negative log-likelihood, i.e.

$$\mathcal{L}_{train} = \mathbb{E}_{\rho \sim \varrho}[-\log p(X_{2.5D}|I)]. \tag{10}$$

However, the camera distribution $\varrho$ and the 2D projection $\Pi_\rho(\mathbf{X}_{3D})$ under each viewport is unknown for each image. In the 2D image dataset $\mathcal{D} = \{\mathbf{I}_1, \mathbf{I}_2, \dots\}$ collected from the Internet, each image has one viewport $\rho$ and one 2D projection $\Pi_\rho(\mathbf{X}_{3D})$ naturally, which is apparently insufficient to train the normalizing flow. To get other views and the corresponding 2D projections for each image, we group these images using the k-nearest neighbor grouping algorithm[8] based on the distance metric defined following

$$d(\mathbf{I}, \mathbf{I}') = \frac{1}{n} \sum_{i=1}^{n} \|(\mathbf{d}_i \times \mathbf{d}_i') \dots (\mathbf{t}_{\text{cam}} - \mathbf{t}_{\text{cam}}')\|, \qquad (11)$$

where $\mathbf{X}_{2.5D} = \left(\mathbf{d}_1^T, \mathbf{d}_2^T, \dots, \mathbf{d}_n^T, \mathbf{t}_{\text{cam}}^T\right)^T$ is the intermediate representation calculated from the 2D keypoints $\Pi_\rho(\mathbf{X}_{3D})$ and the camera pose $\rho = \{\mathbf{R}_{\text{cam}}, \mathbf{t}_{\text{cam}}\}$ for image $\mathbf{I}$ according to the left side of equation (6) and $\mathbf{X}_{2.5D}' = \left(\mathbf{d}_1'^T, \mathbf{d}_2'^T, \dots, \mathbf{d}_n'^T, \mathbf{t}_{\text{cam}}'^T\right)^T$ is the intermediate representation for the image $\mathbf{I}'$. Equation (11) calculates the average distance between two rays which starting from $\mathbf{t}_{\text{cam}}$ and $\mathbf{t}_{\text{cam}}'$ with the direction $\mathbf{d}$ and $\mathbf{d}'$ respectively. The smaller the value of $d(\mathbf{I}, \mathbf{I}')$, the more likely that these rays intersect with each other in 3D space, which indicates that the image $\mathbf{I}$ and $\mathbf{I}'$ capture the same interaction. The grouping process is totally free of human intervention and the algorithm outputs the top-k nearest neighbor for each image $\mathbf{I}$ in the form of cluster

$$\mathcal{G}_{\mathbf{I}} = \{(\Pi_{\rho_1}(\mathbf{X}_{3D}), \rho_1, d_1), \dots, (\Pi_{\rho_k}(\mathbf{X}_{3D}), \rho_k, d_k)\}, \qquad (12)$$

where $d_i$ is the distance between the image $\mathbf{I}$ and the $i$-th item in the cluster $\mathcal{G}_{\mathbf{I}}$. We drop the neighbors whose distance exceeds a threshold. During training, we random select a batch of neighbors from $\mathcal{G}_{\mathbf{I}}$ for each image $\mathbf{I}$ to minimize the loss objective shown in equation (10).

## 3.3 Human-Object Reconstruction with 2D Prior

We consider the task of reconstructing the human and the object from a single-view image with known object templates where the human is parameterized using the shape parameter $\boldsymbol{\beta}$ and pose parameter $\boldsymbol{\theta}$ of SMPL and the object is parameterized by the 6D pose $\{\mathbf{R}, \mathbf{t}\}$ and the scale $s$ of the known object template. We adopt a two-stage prediction-optimization paradigm to recover these parameters $\{\boldsymbol{\beta}, \boldsymbol{\theta}, \mathbf{R}, \mathbf{t}, s\}$ from given the image $\mathbf{I}$, like many existing methods, where these parameters are initialized using the pre-trained pose estimation model, followed by an iterative optimization process to refine the pose of the human and the object.

**Initialization.** We first use the state-of-the-art 3D human mesh recovery model SMPLer-X[3] to predict the shape parameter $\boldsymbol{\beta}$, the pose parameter $\boldsymbol{\theta}$ and the gobal pose $\{\mathbf{R}_{SMPL}, \mathbf{t}_{SMPL}\}$ for SMPL and the pre-trained CDPN [19] to obtain the 6D pose $\{\mathbf{R}, \mathbf{t}\}$ of the object. The scale $s$ of the object template is initialized empirically according to the size of the category in the real world. Besides we extract the keypoints for the human using ViTPose[30] and the 2D-3D corresponding maps using the pre-trained CDPN[19].

**Prior Loss.** The prior learned from 2D images is flexible enough to be deployed to tune the parameter $\boldsymbol{\beta}, \boldsymbol{\theta}$ of SMPL and the 6D pose $\mathbf{R}, \mathbf{t}$ of the object during post-optimization. Given the input image $\mathbf{I}$, we draw $m$ samples from the distribution $p(\Pi_\rho(\mathbf{X}_{3D}), \rho|\mathbf{I})$

learned by the normalizing flow to initialize the translation poses $\{\hat{\mathbf{t}}_{\text{cam}}^{(1)}, \dots, \hat{\mathbf{t}}_{\text{cam}}^{(m)}\}$ for each virtual camera. Then the 3D human-object keypoints $\mathbf{X}_{3D}$ computed from $\{\boldsymbol{\beta}, \boldsymbol{\theta}, \mathbf{R}, \mathbf{t}, s\}$ according to equation (2) and (3) are then projected onto the image planes of each virtual camera to get the intermediate representation $\{\hat{\mathbf{X}}_{2.5D}^{(1)}, \dots, \hat{\mathbf{X}}_{2.5D}^{(m)}\}$ according to the right side of equation (6) and equation (7). The multi-view keypoints prior loss is defined as

$$\mathcal{L}_{\text{prior}} = -\sum_{i=1}^{m} \log p(\hat{\mathbf{X}}_{2.5D}|\mathbf{I}). \qquad (13)$$

The camera translation poses $\{\hat{\mathbf{t}}_{\text{cam}}^{(1)}, \dots, \hat{\mathbf{t}}_{\text{cam}}^{(m)}\}$ are treated as the optimization parameters which are optimized together with $\{\boldsymbol{\beta}, \boldsymbol{\theta}, \mathbf{R}, \mathbf{t}, s\}$ during post-optimization process.

**Contact Loss.** In addition to constraining the 3D human-object keypoints, we also use the contact loss to generate more fine-grained interaction. The contact map is also hard to be acquired in the wild. The contact between the human and the object in 3D space will result in occlusion in the 2D image plane, and inversely, the contact map can be approximated from occlusion in different views. Based on this idea, we approximate it using the average occlusion map. Denote the occlusion map for the human as the binary array $\mathbf{c}_{\text{h}} \in \mathbb{R}^{6890}$ where the element is set to 1 if the corresponding projected coordinate in the image plane falls within the occlusion region with the object. The occlusion map $\mathbf{c}_{\text{o}}$ for the object is defined similarly. For each image from the dataset, we calculate the occlusion maps for the human and the object and average them to get the mean occlusion map $\bar{\mathbf{c}}_{\text{h}}$ and $\bar{\mathbf{c}}_{\text{o}}$. During optimization, we compute the occlusion map for the human and the object from the target image and multiply them with the mean occlusion maps to get the candidate indices set of the points that are under contact. The contact index set for SMPL mesh is given by $C_{\text{h}} = \{i|[\mathbf{c}_{\text{h}}]_i \cdot [\bar{\mathbf{c}}_{\text{h}}]_i > \eta\}$ and the contact index set for object mesh is given by $C_{\text{o}} = \{i|[\mathbf{c}_{\text{o}}]_i \cdot [\bar{\mathbf{c}}_{\text{o}}]_i > \eta\}$, where $\eta$ is contact threshold. The contact loss is defined as the weighted chamfer distance between the two contact point clouds decided by the index set $C_{\text{h}}$ and $C_{\text{o}}$.

$$\mathcal{L}_{\text{contact}} = \frac{1}{|C_{\text{h}}|} \sum_{i \in C_{\text{h}}} \min_{j \in C_{\text{o}}} w_{ij} \|\mathbf{p}_i^{\text{h}} - \mathbf{p}_j^{\text{o}}\| +$$
$$\frac{1}{|C_{\text{o}}|} \sum_{j \in C_{\text{o}}} \min_{i \in C_{\text{h}}} w_{ij} \|\mathbf{p}_i^{\text{h}} - \mathbf{p}_j^{\text{o}}\|, \qquad (14)$$

where $w_{ij} = [\mathbf{c}_{\text{h}} \cdot \bar{\mathbf{c}}_{\text{h}}]_i [\mathbf{c}_{\text{o}} \cdot \bar{\mathbf{c}}_{\text{o}}]_j$, $\mathbf{p}_i^{\text{h}}$ is the $i$-th point in SMPL mesh and $\mathbf{p}_j^{\text{o}}$ is the $j$-th point in the object mesh.

**Optimization Objective.** The overall optimization objective is defined as

$$\mathcal{L}_{\text{optim}} = \lambda_{\text{J}} \mathcal{L}_{\text{J}} + \lambda_{\text{coor}} \mathcal{L}_{\text{coor}} + \lambda_{\text{norm}} \mathcal{L}_{\text{norm}} +$$
$$\lambda_{\text{prior}} \mathcal{L}_{\text{prior}} + \lambda_{\text{contact}} \mathcal{L}_{\text{contact}}, \qquad (15)$$

where $\mathcal{L}_{\text{J}}$ is the reprojection loss for SMPL keypoints, $\mathcal{L}_{\text{coor}}$ is the reprojection loss for the object defined in [13] and $\mathcal{L}_{\text{norm}}$ is the regularization term for the pose of the human and the scale of the object. Our optimization process consists of two phases. In the first phase, we lock the parameters for SMPL and only optimize the 6D pose $\{\mathbf{R}, \mathbf{t}\}$ and the scale $s$ of the object. In the second phase, we tuned all the parameters together by minimizing the optimization objective (15).

# 4 DATASET

In order to validate our method in natural scenes, we collected the WildHOI dataset, which consists of a diverse range of videos from the YouTube website, capturing various natural scenes and human-object interactions.

## 4.1 Data Collection and Preprocessing

Before data collection, we select the object categories from COCO dataset that have almost fixed shapes and cannot be deformed, such as baseball, tennis, basketball, etc. Then, we search for the videos on the YouTube website that contain interactions between humans and these selected object categories. We manually reviewed the videos to ensure they depict the interactions of interest with target object categories. Once we identify relevant videos, we download them and extract frames from these videos. Next, we use bigdetection [2] to extract the bounding boxes of persons and objects in each frame. The person and the object are considered as being under interaction if the IoU of their bounding boxes exceeds a certain threshold. The images without the engagement of any human-object interaction are discarded. After detecting all bounding boxes, we run SAM[17] in each frames to extract the masks within the detected bounding boxes. We also annotate the images with the 2D person keypoints and pseudo SMPL parameters using ViTPose [30] and SMPLer-X[3]. We further tune the SMPL parameters predicted by SMPLer-X using reprojection loss to make the SMPL parameters aligned well with the keypoints extracted by ViTPose.

## 4.2 Human-Object Keypoint Annotations

The keypoints for the human $\Pi_\rho(X_{3D}^{SMPL})$ is easy to be acquired as we can reproject the SMPL joints to each image plane directly. However, due to the diversity of the object categories and the variety of the object shape, there lacks of pretrained models for extracting the object keypoints or estimating the object 6D pose in the wild. To address the problem, we annotate the object keypoints from scratch. We employ multiple annotators to annotate the correspondence between the 2D image coordinates and 3D points on the object template mesh surface. Once the correspondence is obtained, the 6D pose can be calculated using PANSAC/P$n$P algorithms. Annotating all the frames in the dataset is not realistic. To alleviate the annotation workload, we adopt the human-in-the-loop annotation process. We start by randomly selecting a few frames and deliver them to the annotators for labeling. The annotated data is then used to train the 6D pose estimation model. This model is then used to annotate the remaining frames. Human annotators review these annotated images by the pre-trained models and select the incorrect ones. The incorrect images are then handed over to the annotators for correction. The annotated images are used to improve the annotation quality of the 6D pose estimation model iteratively. This iterative annotation process continues until the annotation quality meets our standards. After obtaining the 6D pose annotation for the object, the keypoints for the object $\Pi_\rho(X_{3D}^{object})$ are obtained by projecting the selected points $\hat{X}_{3D}^{object}$ onto the image planes with the estimated 6D pose. The keypoints for the human and the keypoints for the object are concatenated together to get $\Pi_\rho(X_{3D})$, where the camera pose $\rho$ is acquired from the global pose of SMPL. In the end, each image is labeled with the 2D human-object keypoints $\Pi_\rho(X_{3D})$ and the 6D camera pose $\rho = \{R_{cam}, t_{cam}\}$.

## 4.3 Dataset Statistics

Overall, our dataset contains diverse interactions with 8 object categories in various real-world scenarios. Each image is annotated with the bounding boxes, masks, SMPL pseudo parameters, and the human-object keypoints. We split the dataset into training and testing sets with 4:1 ratio, which results in about 30k-100k frames in the training set for each object category. To evaluate our method, we select and annotate a small fraction of images (about 2.5k) from the test set. The pseudo annotations for the poses of the human and the object in the small test set are obtained by optimizing with the contact labels that are manually annotated. For the non-contact interaction types, we ask the annotators to adjust the 6D pose of the object manually in the real-time rendering interfaces.

# 5 EXPERIMENTS

We conduct experiments on both the indoor BEHAVE dataset and the in-the-wild WildHOI dataset.

## 5.1 Experiment on Indoor Dataset

**Dataset and Metrics.** To compare our method with previous 3D supervised methods, we conduct experiments on the BEHAVE dataset[1]. The BEHAVE dataset is a large-scale 3D full-body human-object interaction dataset that contains 8 subjects interacting with 20 common objects. This dataset is captured by four Kinect RGB-D cameras at 30 FPS. Each image is annotated with pseudo SMPL labels, 6D object pose labels, camera poses, and camera calibration parameters. In experiments, we follow the official splits, using 217 video sequences for training and 82 video sequences for testing. To speed up the test process, we only test on the key frames[29]. In terms of evaluation metrics, we use the chamfer distance between the reconstructed 3D mesh with the ground truth. Before calculating the chamfer distance, 10,000 points are sampled from the surface of the mesh. The point clouds sampled from the surface of the reconstructed mesh and the ground-truth mesh are aligned using optimal Procrustes alignment and then the chamfer distance between these two point clouds is calculated. The chamfer distances for the human and the object are reported separately.

**Implementation Details.** Since our method is very sensitive to the accuracy of the 2D human-object keypoints, we use the 2D human-object keypoints rendered from the ground truth and the camera pose obtained from the 3D annotations to train our normalizing flow. Note that we only access the 2D human-object keypoints and the 6D camera pose without directly accessing the SMPL annotations and the object 6D pose labels during the training process of the normalizing flow which doesn't violate the original 2D-supervised goal. The cluster size $k$ is set to 8 in the top-$k$ nearest neighbor grouping. The normalizing flow is trained objectwisely for 30 epochs with the learning rate set to $1e-4$. In the process of inference, we initialize the SMPL parameters and the object 6D pose using ProHMR [18] and Epro-PnP[6] respectively. In post-optimization stage, $\lambda_J, \lambda_{coor}, \lambda_{prior}$ is set to 0.1, 0.1, 1 respectively. The number of

the virtual camera $m$ is set to 8 by default. We didn't use the contact loss $\mathcal{L}_{\text{contact}}$ in the post-optimization process on the BEHAVE dataset.

**Baselines.** We compare our method with previous 3D supervised methods CHORE[28] and StackFLOW[13]. Since our method is supervised with 2D keypoints, it is unfair to compare our method with these methods trained with 3D annotations. We compare our method with theirs to see the marge between the 3D supervised methods and the 2D supervised methods.

**Main Results.** The comparison in terms of reconstruction accuracy between the 3D supervised methods and the 2D supervised methods is shown in table 1. Compared with the 3D supervised methods CHORE and StackFLOW, our method achieves almost comparable performance even if we don't access the 3D annotation directly, which indicates that we find a lighter way to learn the human-object spatial relation prior without using the 3D annotations generated by the calibrated multi-view capture systems.

| Methods | Supervision | SMPL (cm) ↓ | Obj. (cm) ↓ |
|---|---|---|---|
| CHORE[28] | 3D | 5.55 | 10.02 |
| StackFLOW[13] | 3D | 4.79 | **9.12** |
| PHOSA[32] | - | 12.86 | 26.90 |
| Ours | 2D | **4.55** | 11.32 |

Table 1: Performance comparison between the 3D supervised methods and the 2D supervised methods on the BEHAVE dataset.

**Ablation on Different Supervisions.** In table 2, we show the impact of different supervisions on reconstruction accuracy. We train the normalizing flow in three ways: (1) directly using 3D human-object keypoints to train the normalizing flow, (2) training the normalizing flow using 2D human-object keypoints but with the ground truth grouping, (3) training the normalizing flow using 2D human-object keypoints and the grouping is constructed by the KNN algorithm. As shown in table 2, as we expected, the 3D supervision using 3D human-object keypoints (the second line in table 2) achieves the highest reconstruction accuracy. However, using 2D keypoints with ground truth (the third line in table 2) also yields favorable results, closely following the performance of direct 3D supervision. Finally, using 2D keypoints with grouping from the KNN algorithm (the last line in table 2) results in slightly lower accuracy but still performs reasonably well. This indicates that even without direct access to 3D annotations, it is possible to achieve comparable performance with a minimal drop in accuracy by utilizing 2D keypoints and appropriate grouping strategies.

| Supervision | Grouping | SMPL (cm) ↓ | Obj. (cm) ↓ |
|---|---|---|---|
| 3D | - | 4.34 | 10.23 |
| 2D | GT | 4.51 | 10.99 |
| | KNN | 4.55 | 11.32 |

Table 2: The effectiveness of different supervisions and KNN grouping on the reconstruction accuracy.

## 5.2 Experiment on Outdoor Dataset

**Dataset and Metrics.** The WildHOI dataset is used to evaluate the performance of our method on in-the-wild images. We evaluate our method using the chamfer distance but with a slight difference from the evaluation metric on the BEHAVE dataset. The reconstruction meshes and the ground-truth meshes are placed on the local coordinate system of SMPL where the pelvis joint of SMPL is rooted at the origin. The chamfer distance between the reconstruction mesh and the ground-truth mesh is calculated without using the optimal Procrustes alignment. Besides, we also use the rotation error and the translation error to evaluate the relative pose between the object and the human. In addition to the numerical results, we also conduct human evaluation. More details about human evaluation can be found in the supplementary materials.

**Implementation Details.** We employ the normalizing flow with a depth of 8 and a width of 512. The training time of the normalizing flow on each object category varies depending on the convergence of its loss. The cluster size $k$ is set to 16 in the top-$k$ nearest neighbor grouping algorithm. During post-optimization, $\lambda_J$ is set to 0.01, while $\lambda_{\text{prior}}$ and $\lambda_{\text{coor}}$ are set to 0.1, $\lambda_{\text{contact}}$ is set to 1. We use the corresponding maps generated using 6D pose labels to calculate the loss $\mathcal{L}_{\text{coor}}$ in equation (15).

**Baselines.** Most recent works are learning-based methods that require 3D annotations for training. Compared with them our 3D-free method will be unfair since we don't have large-scale 3D annotations for in-the-wild images. We have to select the annotation-free method PHOSA[32] as our main baseline. To make the comparison fairer, we adapt the PHOSA into our method by substituting the loss $\mathcal{L}_{\text{prior}}$ and $\mathcal{L}_{\text{contact}}$ in equation (15) with the coarse interaction loss $\mathcal{L}_{\text{coarse inter}}$, the fine interaction loss $\mathcal{L}_{\text{fine inter}}$ and the ordinal depth loss $\mathcal{L}_{\text{depth}}$ proposed in PHOSA with all the other settings the same with our methods.

**Main Results.** As shown in table 3, our method outperforms PHOSA in terms of all evaluation metrics, especially in terms of the translation error of the object. Because PHOSA and our method all used the same SMPL parameters predicted by SMPLer-X and the same correspondence maps of the object rendered from the object 6D pose labels, there is only a slight difference in chamfer distance of SMPL and the rotation error of the object. The improved performance of our method can be contributed by the human-object prior loss $\mathcal{L}_{\text{prior}}$. With this strong prior learned from vast 2D images, our method leads to lower translation error on objects and better overall performance compared to PHOSA.

| Methods | SMPL(cm)↓ | Obj.(cm)↓ | Rot.(°)↓ | Transl.(cm)↓ |
|---|---|---|---|---|
| PHOSA | 4.72 | 50.08 | 11.90 | 33.22 |
| Ours | **4.43** | **17.48** | **10.12** | **13.13** |

Table 3: The comparison between PHOSA and ours on the WildHOI dataset.

**Qualitative Results.** As shown in figure 3, we compare the qualitative results of our method with PHOSA. From the qualitative results, we can see that our method can accurately reconstruct the spatial relation between the human and the object in different scenarios. Although the reconstruction results of PHOSA can align well with the image, the reconstruction is not coherent when observed from

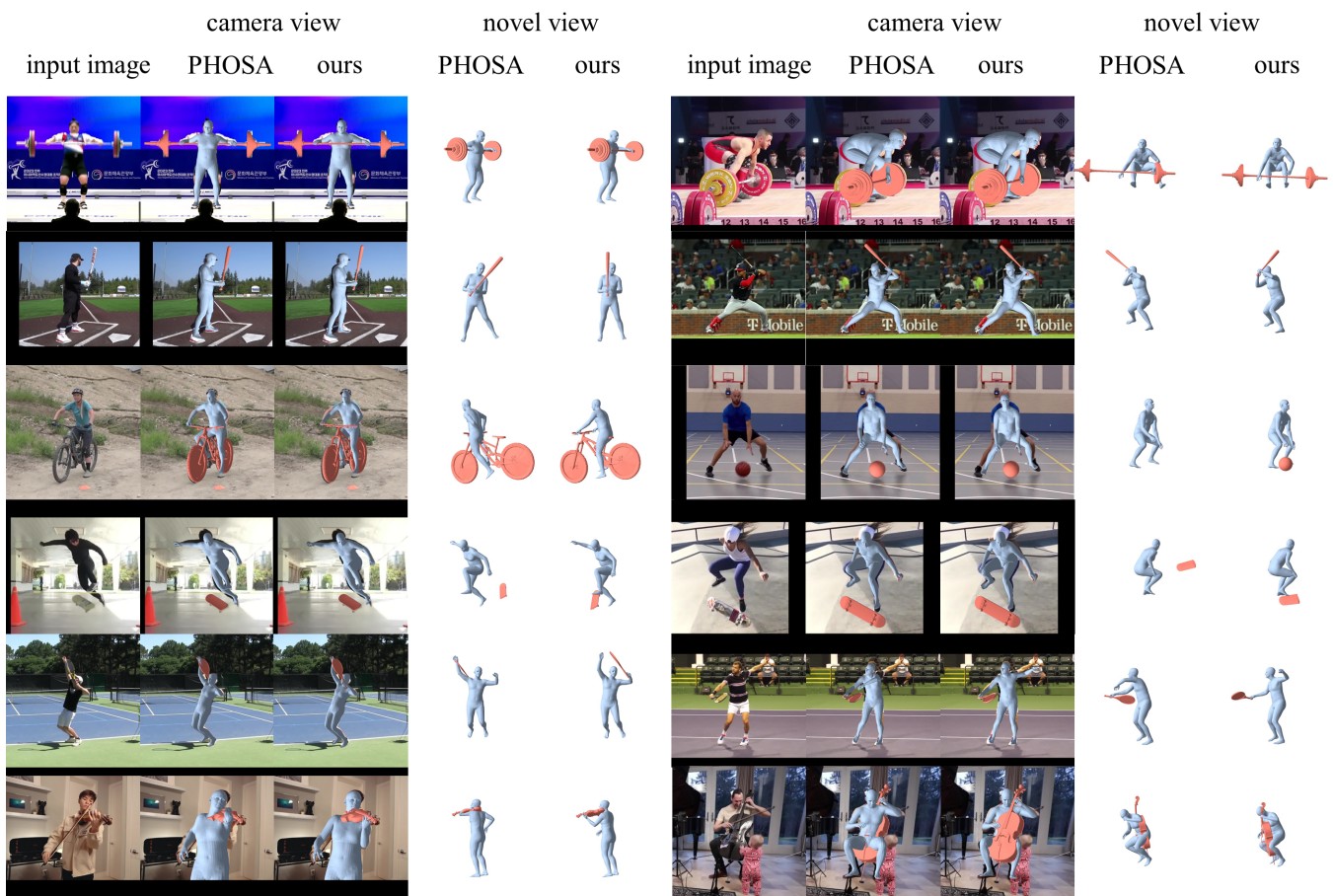

**Figure 3: The qualitative results on WildHOI dataset.**

the side view. Moreover, our method can deal with non-contact interaction types, whereas PHOSA, which relies on the contact map to constrain the relative pose between the human and the object, fails in such cases of non-contact interaction.

**Ablation on the Optimization Loss.** Table 4 shows the results of ablation experiments on different optimization losses. From the results, we can see that including both the prior and contact losses leads to the best performance, achieving the lowest errors in all metrics except the chamfer distance for SMPL. This indicates that both the prior and contact losses play important roles in the optimization process. By comparing the third line and the last line, we can see that excluding the prior loss leads to a significant drop, which indicates that the prior loss has a more significant impact on reconstruction accuracy.

| $\mathcal{L}_{prior}$ | $\mathcal{L}_{contact}$ | SMPL(cm)↓ | Obj.(cm)↓ | Rot.(°)↓ | Transl.(cm)↓ |
|---|---|---|---|---|---|
| ✗ | ✗ | **3.57** | 1259.57 | 7.12 | 658.16 |
| ✗ | ✓ | 3.71 | 363.40 | 6.47 | 195.62 |
| ✓ | ✗ | 4.36 | 19.37 | 10.26 | 14.26 |
| ✓ | ✓ | 4.43 | **17.48** | **10.12** | **13.13** |

**Table 4: The effectiveness of different losses on the reconstruction accuracy.**

## 6 CONCLUSION

In this work, we explore how to learn strong prior of the spatial relation between the human and the object from 2D images in the wild. Through our experiments, we have shown that, even without using any 3D annotations or commonsense knowledge of the 3D spatial relation between the human and the object, our method can achieve impressive results on both the indoor BEHAVE dataset and the outdoor WildHOI dataset. However, there are still some limitations in our work. First, our method only focuses on learning the 3D spatial relation prior between the human and the object with the assumption that the shape of the object is known. This may not be practical in real-world scenarios where the object shapes can vary greatly. Additionally, our method relies heavily on the availability of large-scale 2D image datasets, which may not always be readily available or applicable to all tasks. Furthermore, our method learns the instance-level prior rather than the category-level prior. This may affect the generalization ability to unseen or rare object categories.

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
