# OpenReview forum: "Monocular Human-Object Reconstruction in the Wild"
_acmmm.org/ACMMM/2024/Conference — MM2024 Poster_

### Official Review · Reviewer_etVg · 2024-05-23

**Rating:** 4
**Confidence:** 2

**Summary:**

This paper focuses on reconstructing 3D human and object interaction from in-the-wild RGB images. It has two stages for reconstruction. First, it proposed to learn the interaction prior from 2D keypoints layouts and then use this prior to benefit the following optimization process. It can learn the prior without using 3D annotations and achieve good performance.

**Strengths:**

- The paper proposes a novel method to learn the prior from 2D keypoints layouts for humans and objects.
- It proposes an optimization pipeline based on learned priors to perform the reconstruction.
- It introduces a wild datased called WildHOI that can better benchmark method in in-the-wild domains.

**Limitations:**

- What's is the motivation of using normalizing flows model to learn the prior? Would VAE or diffusion model be a better choice for this case?
- Why the in-the-wild generalization an important issue for this problem? Since most methods in this track build upon optimization, it is feaible to first use foundation models to get segmentation masks, keypoints and use optimization to align 3D with these 2D evidence. Therefore, I think optimization-based methods can easily work on these in-the-wild scenes.
- Lacking of comparison with baselines. The only baseline compared is PHOSA. Actually, there are other works [a, b] that need to be compared with, which could really show the advantages of the proposed method.

[a] Chore: Contact, human and object reconstruction from a single rgb image
[b] Reconstructing Action-Conditioned Human-Object Interactions Using Commonsense Knowledge Priors

**Suitability:**

2

---

### Official Review · Reviewer_82xq · 2024-05-26

**Rating:** 5
**Confidence:** 3

**Summary:**

This paper proposes a method to learn the prior knowledge of the 3D human-object spatial relation from 2D images in the wild.
Unlike existing methods that require 3D datasets with laborious annotations, the proposed method stands out for its practicality, only necessitating 2D annotated data through the use of Normalizing Flow. The method is rigorously evaluated using the newly introduced dataset called WildHOI, and it demonstrates promising results, outperforming baseline methods.

**Strengths:**

- The paper is carefully written, and the implementation of the method is discussed in detail.
- The proposed method introduces a high level of novelty. As detailed in the Related Work section, while some studies require 2D annotated data for human pose estimation, 3D scene reconstruction, or hand-object reconstruction, the proposed method in this paper takes a unique approach by using Normalizing Flow to represent the distribution of 2D human-object keypoints for the prior of human-object relation.
- The proposed method has been evaluated on the human-object reconstruction task. In this evaluation, the paper successfully demonstrates the contribution of the proposed prior by showing a significantly reduced translation error of the reconstructed objects.
- This study also provides a newly constructed WildHOI dataset collected from YouTube websites, capturing various natural scenes and human-object interactions. Together with the annotation data included, the dataset may have a significant impact on the multimedia community, especially in human-object studies.

**Limitations:**

- As discussed in the Conclusion section, the proposed method assumes that the object's shape is known. In addition, the method still requires the annotation of keypoints of the objects. It is not clear how the method can be extended to unknown objects or various object types in the wild (i.e., collected from the Internet).
- While the concept of contact was already utilized as the occlusion-aware loss in [32], this fact is only mentioned in the last part of Section 5 (line 849). It would be helpful for readers to explain the originality of the proposed contact loss compared to [32] in (or before) Section 3.3.

**Suitability:**

3

---

### Official Review · Reviewer_ej4a · 2024-05-27

**Rating:** 4
**Confidence:** 4

**Summary:**

This paper presents a 2D-supervised method that learns the 3D human-object spatial relation prior purely from 2D images in the wild. The method utilizes a flow-based neural network to learn the prior distribution of the 2D human-object keypoint layout and viewports for each image in the dataset. The effectiveness of the prior learned from 2D images is demonstrated on the human-object reconstruction task by applying the prior to tune the relative pose between the human and the object during the post-optimization stage. To validate and benchmark method on in-the-wild images, the authors collect the WildHOI dataset from the YouTube website, which consists of various interactions with 8 objects in real-world scenarios. The results show that proposed method achieves almost comparable performance with fully 3D supervised methods on the BEHAVE dataset, even if only utilized the 2D layout information, and outperforms previous methods in terms of generality and interaction diversity on in-the-wild images.

**Strengths:**

1.The paper is mostly well-written and easy to follow.

2.This paper presents a 2D-supervised method that learns the 3D human-object spatial relation prior purely from 2D images in the wild.

3.This paper is informative and well experimented.

4.This paper proposes a novel dataset that is a positive contribution to the development of this field.

**Limitations:**

Overall this work meets the ACM MM standard, but there are still some concerns regarding the dataset. 3D human-object interaction annotation is and its difficult, and in this paper only some indirect means are used to ensure the accuracy of the GTs, and the GTs obtained in this way are actually in error. Although I agree with the importance of this dataset, the error in GT leads to the same bias in the trained model, which will directly affect the results in downstream applications.

**Suitability:**

3

---

### Meta-Review · Area_Chair_SwNi · 2024-07-01

**Recommendation:** Accept (Poster)
**Confidence:** 4

**Metareview:**

This paper presents a 2D-supervised method that learns the 3D human-object spatial relation prior from 2D images in the wild. The method utilizes a flow-based neural network to learn the prior distribution of the 2D human-object keypoint layout and viewports for each image in the dataset. The effectiveness of the prior learned from 2D images is demonstrated on the human-object reconstruction task by applying the prior to tune the relative pose between the human and the object during the post-optimization stage. To validate and benchmark method on in-the-wild images, the authors collect the WildHOI dataset from the YouTube website, which consists of various interactions with 8 objects in real-world scenarios. The results show that proposed method achieves almost comparable performance with fully 3D supervised methods on the BEHAVE dataset, even if only utilized the 2D layout information, and outperforms previous methods in terms of generality and interaction diversity on in-the-wild images.

After the rebuttal, all the reviewers are positive about the paper.